materials science/environmental chemistry

hierarchical AlPO-34, ionothermal synthesis, self-assemble, heavy metal adsorption

**Author for correspondence:**
Runlin Han
e-mail: hanrunlin@163.com

# Synthesis of self-assembled hierarchical AlPO-34 microspheres by using an ionic liquid and its application in heavy metal removal

Liang Zhou[1], Hongjian Li[1] and Runlin Han[1,2]

[1]School of Chemical Engineering, Dalian University of Technology, Dalian 116024, People's Republic of China
[2]School of Chemistry and Chemical Engineering, Jinggangshan University, Ji'an 343009, People's Republic of China

RH, 0000-0002-4992-5149

Hierarchical AlPO-34 molecular sieves microspheres were synthesized in a [BMIm]Br ionic liquid without template or complex post-treatment process. The formation mechanism of such framework structures and their morphology were investigated. [BMIm]Br was proven to serve as both solvent and sole provider of the structure directing agent. The organic amine in the compound affects the framework density of the crystals and promotes the formation of a chabazite (CHA) type framework. After ageing for 1 h the AlPO-34 microspheres are formed due to the aggregation properties of the ionic liquid. The hierarchical mirosphere has a relatively high Brunauer–Emmett–Teller surface area and a considerably uniform mesoporous channel network. The hierarchical AlPO-34 microspheres were used as absorbers of heavy metal cations and showed a higher loading capacity and distribution coefficient compared with the AlPO-34-NH.

## 1. Introduction

Zeolitic materials are one of the oldest and the most accessible porous crystalline compounds which are widely used in many fields, such as catalysis, gas separation, adsorption and wastewater treatment [1]. Their unique channels often provide a relatively efficient performance. However, the low surface area and small micropores size (usually below 2 nm) limit their properties to some extent [2]. In order to overcome such intrinsic limitations and enhance the performance of conventional zeolites, recent investigations mostly focused on providing a solution for the drawbacks [3]. Hierarchically porous spheres have triggered

considerable attention due to their excellent performance when used in drug storage and release, catalysis, coatings and as absorbers [4]. Aluminophosphates (AlPO-n) and isomorphous substituted aluminophosphates with various metals (SAPO-n, MeAPO-n) are studied in detail due to high porosity, good specific area, high thermal and chemical stability [5,6].

Several approaches are employed to obtain spherical materials with hierarchical structures, for instance, the sol-gel method and the template method with a layer-by-layer assembly [7,8]. The self-assembly method was initial reported by Caruso *et al.* to obtain hollow silica and silica-polymer spheres by consecutively assembling silica nanoparticles and polymers onto colloids. The templates were subsequently removed by calcination or decomposition upon exposure to a solvent [9]. However, the syntheses of AlPO-n and SAPO-n often need high temperature which increases the risk and cost of synthesis process [5].

A novel zeolitic molecular sieves synthesis method, known as ionothermal synthesis method, enables the fabrication of zeolites via atmospheric pressure synthesis, due to the negligible vapour pressure of the ionic liquid solvents [10–12]. In this technique, ionic liquids with a low melting point (less than 100°C), high polarity, electric conductivity and thermal stability, and low vapour pressure are used as solvents during the fabrication of the porous materials [13,14]. Interestingly, ionic liquids based on 1-alkyl-3-methylimidazolium cations may show an aggregation behaviour and form micelles in solutions [15], which can be used as core templates for the fabrication of hierarchical microspheres [16].

Adsorption is considered as the one of the best techniques in waste water treatment and gas separation due to the simplicity in set-up and cost efficiency [17,18]. Porous materials with high surface area and strong binding affinity are very promising candidates for sustainable and clean water supply in the future [19,20]. It has been demonstrated that the transition metals have obvious effect on the AlPO-5 zeolite adsorption capacity and the ZnAPO-5 showed higher adsorption capacity for all xylene isomers [21]. FAU type zeolites prepared from coal fly ash were reported for the simultaneous removal of Cd(II), Co(II), Cu(II), Pb(II) and Zn(II) ions from aqueous solutions [22]. The existence of the bridging hydroxyl group in the SAPO structure causes chemisorption of ions which can greatly improve the performance of the adsorbent. AlPO-5 and SAPO-5 were used as novel adsorbents in heavy metal adsorption which showed great potential in removal of heavy metal ions from water [23].

AlPO-34 with a regular block-like crystal morphology is more suitable than many other molecular sieves to prepare via the ionothermal synthesis method [24]. In this work, 1-butyl-3-methylimidazolium bromide ([BMIm]Br) was employed as both the structure agent template and the core template for the fabrication of AlPO-34 molecular sieves. Ageing and time-dependent experiments were carried out to investigate the formation mechanism of the reaction products via the layer-by-layer (LBL) assembly process in ionic liquids. The results show that a hierarchically spherical structure with uniform size can be obtained. Adsorption performance of AlPO-34 was studied to remove heavy metal ions such as Cr(III), Zn(II), Pb(II) and Cu(II) from water.

# 2. Experimental

## 2.1. Synthesis

Initially, [BMIm]Br (Aladdin, 97%), $H_3PO_4$ (85% in water, Aladdin, AR), and aluminiumisopropoxide (Aladdin, AR) were successively added into a beaker and stirred until aluminiumisopropoxide completely dissolved. Then, hydrofluoric acid (40 wt% in water, Sinopharm Chemical Reagent Co., Ltd, AR) and cyclohexylamine (Aladdin, 98%) were slowly added to the solution. The mixture was rapidly stirred (stirring speed of 1000 r.p.m.) for a certain time and transferred into a microwave reaction system (MWD-520, Shanghai Sineo Microwave Chemistry Technology). The compound was heated up with a heating rate of $20°C\,min^{-1}$ and a heating power of 500 kW to foster the crystallization process. To clarify the formation mechanism of the target products, the quantity of organic template and the ageing time of the compound were adjusted. Effect of the hydrothermal synthesis temperature and the reaction time were also investigated. Details about the initial reaction mixture composition, the operating conditions, and the structure of the final products are listed in table 1. After the crystallization, the samples were washed with distilled water and acetone several times via ultrasonic oscillation and then dried overnight at 80°C.

## 2.2. Adsorption experiments

A series of heavy metal solutions was prepared by dissolving analytical grade $Cr(NO_3)_3$, $Cu(NO_3)_2$, $Pb(NO_3)_2$ and $Zn(NO_3)_2$ in deionized water to obtain the 200, 600, 1000, 1400 and 1800 ppb solutions, respectively. An amount of 0.05 g of calcined AlPO-34 was added into a 50 ml solution with different

**Table 1.** Ionothermal synthesis conditions and their corresponding product phases.

| sample[a] | xcyclohexylamine | temperature (°C) | t (min) | ageing time (min) | product |
|---|---|---|---|---|---|
| S1 | 0 | 200 | 120 | 60 | AEL + AFI |
| S2 | 2 | 200 | 120 | 60 | CHA + AEL + AFI |
| S3 | 3 | 200 | 120 | 60 | CHA + AFI |
| S4 | 4 | 200 | 120 | 60 | CHA |
| S5 | 4 | 160 | 120 | 60 | amorphous |
| S6 | 4 | 180 | 120 | 60 | CHA |
| S7 | 4 | 200 | 1 | 60 | amorphous |
| S8 | 4 | 200 | 5 | 60 | CHA |
| S9 | 4 | 200 | 15 | 60 | CHA |
| S10 | 4 | 200 | 30 | 60 | CHA |
| S11 | 4 | 200 | 60 | 60 | CHA |
| S12 | 4 | 200 | 180 | 60 | CHA |
| S13 | 4 | 200 | 120 | 0 | CHA |
| S14 | 4 | 200 | 120 | 30 | CHA |

[a]The initial molar composition for the inorganic species is as follows: $P_2O_5/Al_2O_3/HF/cyclohexylamine/[BMIm]Br = 1/1/2/x/60$.

concentrations of heavy metal ions. The mixtures were stirred for 4 h at ambient temperature. This process was followed by centrifugation to obtain a clear solution. Inductive coupled plasma emission spectrometry (ICP) was employed to measure the concentration of the cations in the initial solution and in the product solution. A series of adsorption experiments was carried out on each solution with different initial concentrations of metal ions. Each measurement was carried out three times in the same conditions. The distribution coefficient ($K_d$, $ml^{-1} g^{-1}$) is a parameter which describes the adsorption behaviour of the heavy metal cations in aqueous solutions. Moreover, it is used to measure the transfer ability of the metal cations from the solutions to the adsorbent. Equation (2.1) was used to determine the removal efficiency while equation (2.2) was used to determine the affinity of the absorbent towards metal ions. $R$ and $K_d$ were calculated by using the following equations:

$$R = \frac{100(C_i - C_f)}{C_i}\%$$ (2.1)

and

$$K_d = \frac{(C_i - C_e)}{C_e} \times \frac{V}{m},$$ (2.2)

where $C_i$, $C_f$ and $C_e$ are the initial, final and equilibrium concentrations of the metal ions in the solution ($mg\, l^{-1}$), respectively. $V$ corresponds to the solution volume (in ml) and $m$ refers to the mass of the adsorbent (in g).

For comparison, AlPO-34 molecular sieves without hierarchical structure (named as AlPO-34-NH) are synthesized according to the previously reported method [25], and used as the adsorbent of heavy metal cations under same experiment conditions with sample S4.

## 2.3. Characterization

The phase purity and the crystallinity of the samples were obtained via powder X-ray diffraction analysis on a Shimadzu XRD-7000S diffractometer fitted with Cu Kα radiation ($\lambda = 1.5418$ Å) operating at 40 mA and 40 kV. The morphology of the particles was analysed via scanning electron microscopy (SEM, Nova Nano SEM 450) and transmission electron microscopy (TEM, Tecnai F30, FEI). Moreover, the textural properties of the calcined samples were determined via nitrogen physisorption at 77 K by using an Autosorb-iQ-C adsorption analyser (Quantachrome Instruments). The surface area was calculated by employing the Brunauer–Emmett–Teller (BET) method. A series of $^{13}$C solid-state cross polarization magic-angle spinning (CP-MAS) NMR measurements was conducted on a Bruker Ultrashield 500

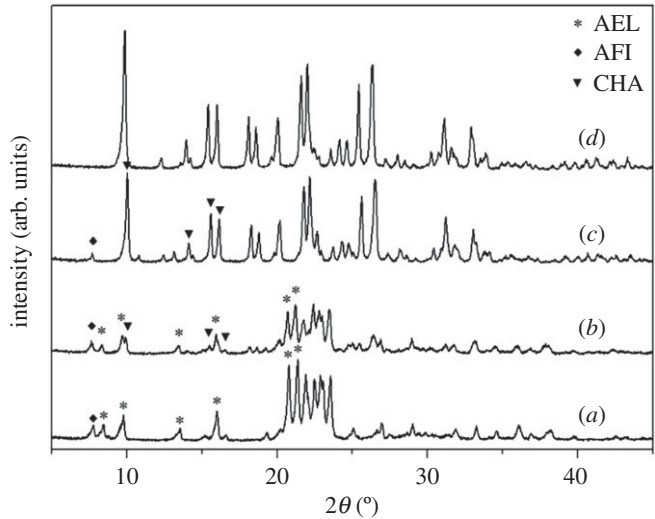

**Figure 1.** XRD patterns of synthetic samples with different cyclohexylamine additions. (*a*) Sample-S1; (*b*) Sample-S2; (*c*) Sample-S3; (*d*) Sample-S4.

spectrometer at the MAS frequencies. The $^{13}C$ CP/MAS NMR spectra were recorded by collecting 2000 scans at a spinning rate of 10 kHz with a contact time of 4 ms and a recycle delay of 4 s. The chemical shifts were referenced to adamantane with the upfield methine peak set to 29.5 ppm. A thermogravimetric analysis (TGA) was performed by using a Mettler-Toledo TGA/DSC1-MS thermal analyser with a heating rate of 10°C min$^{-1}$ in air.

# 3. Results and discussion

## 3.1. Effect of the amount of cyclohexylamine on the sample structure

During the ionothermal synthesis process, the organic amines usually adjust the physico-chemical properties of the solvents and play the role of structural or co-structural directing agents [26]. In this work, cyclohexylamine was introduced into the initial specimen since this organic amine component has been barely employed in previous ionothermal syntheses of CHA type molecular sieves.

Figure 1 shows the X-ray power diffraction (XRD) patterns of different samples (S1–4) prepared by using various amounts of cyclohexylamine. Several diffraction peaks in figure 1*a* can be attributed to the AEL and AFI phases. This implies that the molecular sieves which exhibit the AEL and AFI phases constitute the main reaction products of the ionothermal system, when no cyclohexylamine is added. These results are in agreement with previous reports, which show that [BMIm]Br was used as both the solvent and the structure directing agent without the addition of organic amines. However, a slight decline tendency can be observed from the data reported in figure 1*b*: when the proportion between cyclohexylamine and $Al_2O_3$ is equal to 2, several peaks, which belong to the CHA phase. This indicates that upon the addition of cyclohexylamine, CHA type molecular sieves gradually form. By further enhancing this ratio to 3, the CHA type phase becomes the major reactant product and the amount of AFI phase is extremely limited. Moreover, the diffraction peaks characteristic of the AEL type molecular sieve completely disappear. Further investigations show that an AlPO-34 sample with a CHA structure exhibits a well-defined crystallinity when the ratio between cyclohexylamine and the $Al_2O_3$ source corresponds to 4. Upon the increase of the amount of cyclohexylamine, the AEL and AFI phases are inhibited successively, whereas the formation of the CHA type molecular sieves is promoted. From these results, one can conclude that the cyclohexylamine concentration in the synthesis gel has a remarkable effect on the phase selectivity during the crystallization reaction.

Figure 2 shows the SEM results of the S1–4 crystalline products. The crystals were synthesized without the addition of cyclohexylamine and they present a rod-like shape with a length of about 5 µm and a needle-like shape with a length of about 10 µm (figure 2*a*). According to previous reports, such rod-like products may belong to the AEL-type molecular sieves, whereas the AFI products present a needle-like shape. When the ratio between cyclohexylamine and Al measures 2, a small amount of bulky crystals can be observed in addition to the rod- and needle-like crystals (figure 2*b*).

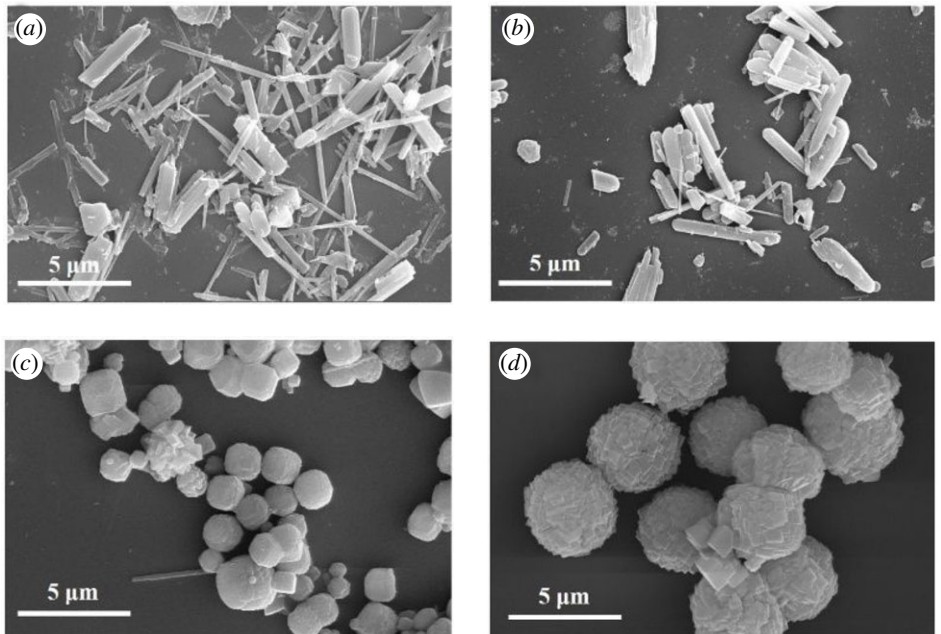

**Figure 2.** SEM images of synthetic samples with different cyclohexylamine additions. (*a*) Sample-S1; (*b*) Sample-S2; (*c*) Sample-S3; (*d*) Sample-S4.

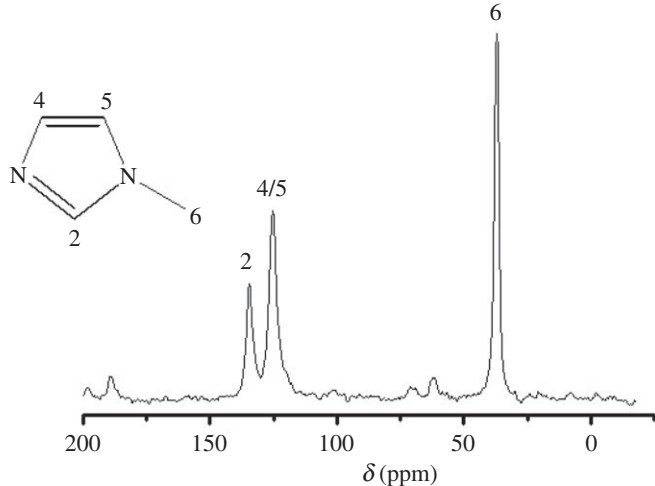

**Figure 3.** $^{13}$C CPMAS NMR spectrum of the S4 sample.

According to their XRD pattern (figure 1*b*), these crystals can be attributed to the presence of the CHA molecular sieves. The majority of crystals present in the S3 sample exhibit a cubic morphology and they have a 2 µm size (figure 2*c*). However, a few crystals still exhibit a needle-like morphology. Figure 2 shows that the amount of AEL and AFI type products decreases or even disappears upon the increase in the cyclohexylamine concentration. When the ratio cyclohexylamine : Al$_2$O$_3$ reaches 4, the products consist of agglomerates of microspheres with a size of approximately 4 µm, which are composed of well-crystallized rhombohedral CHA crystals.

In order to deeply investigate the role of cyclohexylamine and [BMIm]Br in the ionothermal synthesis method, where the CHA type are the molecular sieves, $^{13}$C MAS NMR and TGA were used to characterize the S4 product. As shown in figure 3, three resonances, which are located at $\delta = 37.5$, 75 and 130 ppm, are assigned to *N*-methylimidazole. Moreover, there is no evidence of cyclohexylamine resonances in the $^{13}$C MAS NMR spectrum of AlPO-34, implying that *N*-methylimidazole is the only organic species in the sample. The same conclusions can also be drawn from the TGA curve of the S4 product. A two-step weight loss can be observed in the 30–300°C and 300–600°C temperature ranges

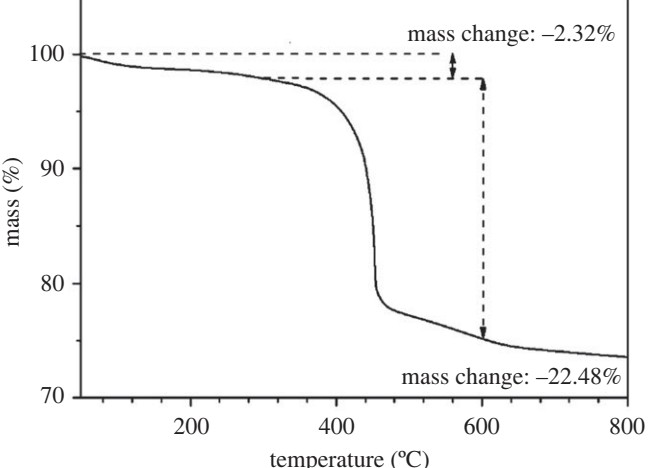

**Figure 4.** Thermogravimetric (TG) curve of the S4 sample.

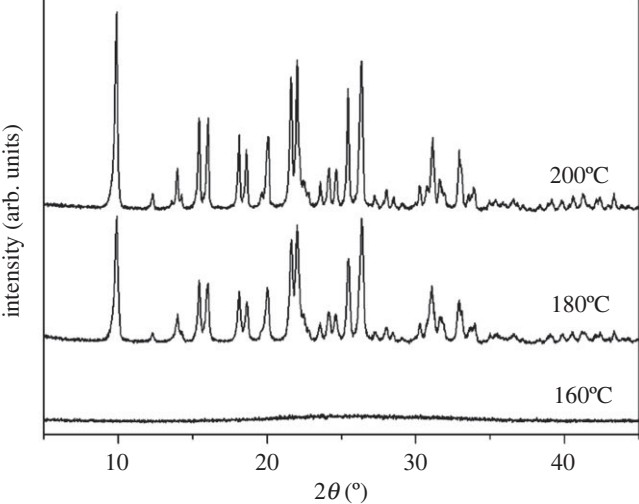

**Figure 5.** XRD patterns of synthetic samples prepared at different temperatures.

for S4 (figure 4). The first one (2.32 wt% weight loss) is attributed to physisorbed water. The remaining mass loss, which occurs between 300°C and 600°C, and measures 22.48% of the total mass corresponds to the loss of *N*-methylimidazole and balances the negative charge of the framework by acting as a pore filler. Moreover, in the presence of a fluoride, some ionic liquids, which present an alkyl chain longer than that of 1-methyl-3-alkylimidazolium bromide, are decomposed into *N*-methylimidazole and methyl bromide. Therefore, 1-butyl-3-methylimidazolium bromide is used as both the solvent and the sole provider of the structure directing agent. These results show that the ionothermal synthesized products, which have a low framework density (FD) structure, can be obtained by increasing the amine concentration in the ionothermal system, leading to the reassembling of the inorganic hosts around the mutated organic guests.

## 3.2. Effects of the crystallization temperature on the sample structure

The crystallization temperature is one of the most important factors to control the purity and the crystallinity of the molecular sieves. Figure 5 shows the XRD patterns of the products synthesized at different crystallization temperatures. The product S5 is amorphous when the crystallization process occurs at 160°C. The CHA phase can be successfully synthesized at the temperature of 180°C or 200°C. Moreover, the product synthesized at 200°C (S4) shows a higher diffraction peak than S5, which confirms that higher temperatures promote the crystallization process.

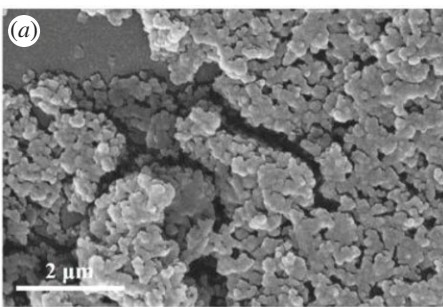
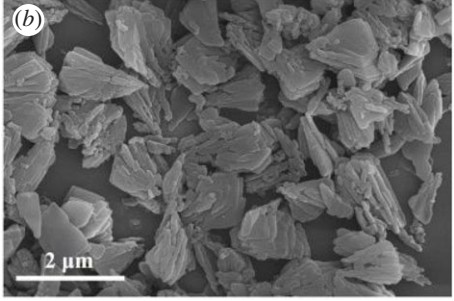

**Figure 6.** SEM images of synthetic samples prepared at different temperatures. Sample S5 (*a*); Sample S6 (*b*).

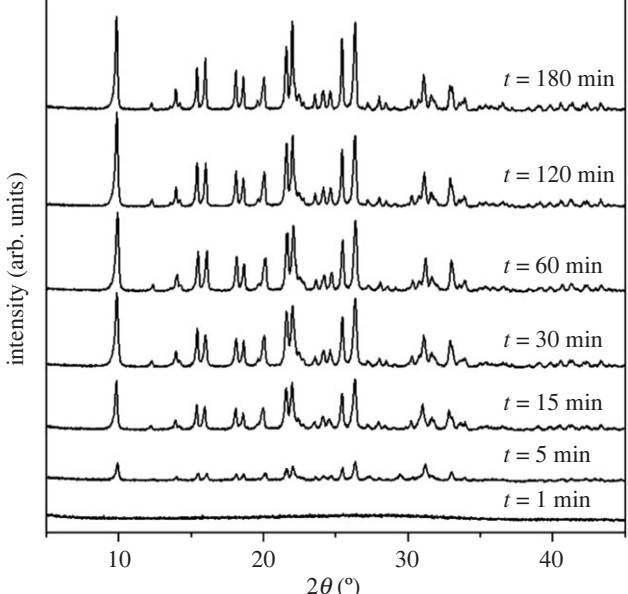

**Figure 7.** XRD patterns of synthetic samples prepared at different time.

Similar results can also be observed in the SEM image of the samples (figure 6): the product of S5 shows a large number of dense phases, whereas the product of S6, which was crystallized at 180°C, shows a series of aggregates consisting of discrete particulates with no obvious crystal edges and corners. As for the product of S4, numerous microspheres with a uniform size can be observed. Such spheres were generated by the assembly of crystals with a distinct CHA morphology. The particle sizes and crystallinity of the products were increased with increase of treatment temperature. According to the XRD of the products, sample S4 prepared with high temperature showed high crystallinity. Therefore, a product with a well-crystallized CHA morphology can be obtained when the crystallization temperature is equal to 200°C.

## 3.3. Effects of the crystallization time on the sample structure

The crystallization time is another key factor in the preparation of molecular sieves. For this reason, it is of utmost importance to investigate the formation mechanism of the molecular microspheres by using comparative time-dependent experiments. In this work, the synthesis process was carried out at 200°C in a time frame of 1–180 min.

Figure 7 presents the XRD patterns of the samples synthesized by using different reaction times. After 1 min, no diffraction peak is observed, indicating that the molecular sieves crystals have not yet been formed. Several weak diffraction peaks which suggest the presence of the CHA topological structure emerge after 5 min in S8. This indicates the generation of CHA molecular sieves. The intensity of the diffraction peaks increases dramatically over the first 2 h of crystallization. After 120 min, no obvious change appears either in the diffraction peaks or in the half-peak width and this demonstrates that the

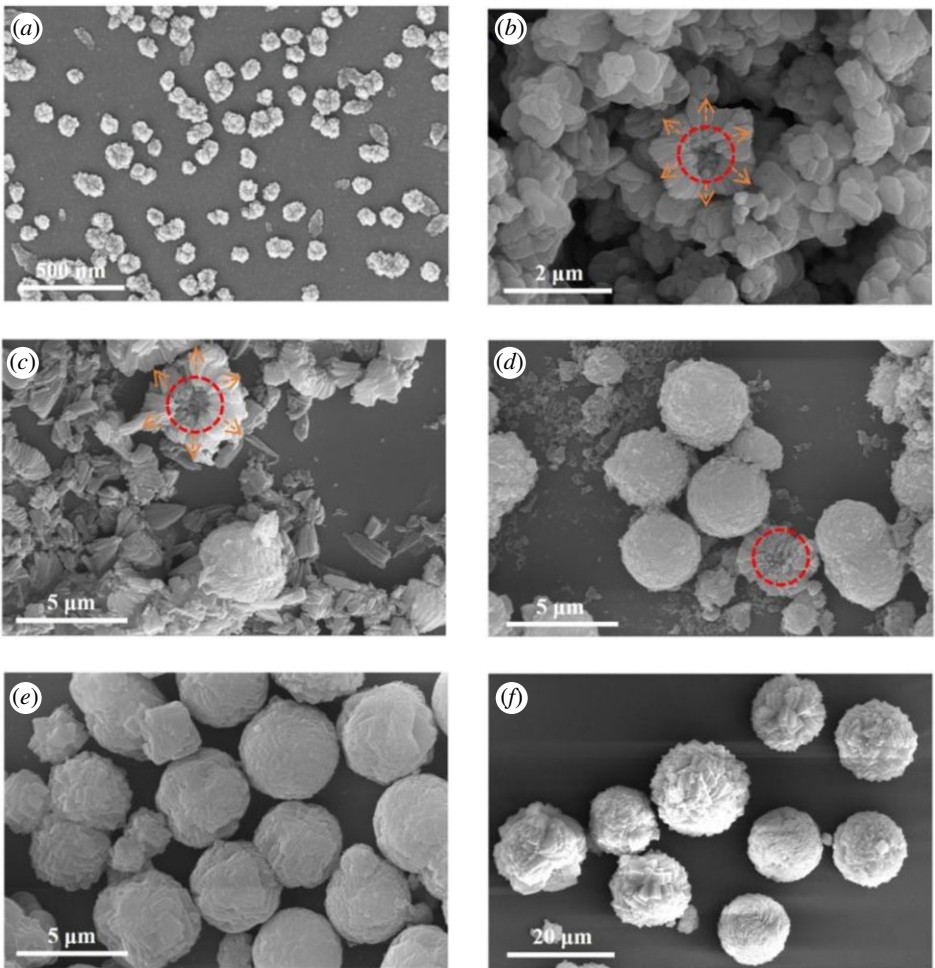

**Figure 8.** SEM images of synthetic samples prepared at different times. (*a*) *t* = 1 min; (*b*) *t* = 5 min; (*c*) *t* = 15 min; (*d*) *t* = 30 min; (*e*) *t* = 60 min; (*f*) *t* = 180 min.

sample crystallizes completely in 120 min. The hydrothermal synthesis of AlPO-34 needs a minimum of 12 h to reach the same crystallinity as CHA, indicating that the ionothermal synthesis significantly shortens the crystallization time and enhances the efficiency in the preparation of the molecular sieves. Moreover, the completion of the crystallization reaction in such a short period of time benefits from the excellent absorption ability of the ionic liquids for microwaves. At the first step, the crystal formed and grew gradually. After 120 min, no obvious change appears either in the diffraction peaks or in the half-peak width and this demonstrates that the sample crystallizes completely in 120 min. The ionothermal synthesis significantly shortens the crystallization time and enhances the efficiency in the preparation of the molecular sieves.

It has been observed that the particle size gradually increased as the synthesis time prolonged, indicating that the particle size was greatly affected by the synthesis time. The evolution of the morphology and size of the products for different crystallization times are displayed in figure 8. In its initial stage, S7 (figure 8*a*) exhibits a dense phase, which is generated by the aggregation of particles with a uniform size of about 100 nm. When the crystallization process time is extended to 5 min and the sample is maintained at 200°C temperature, the morphology of the products changes into cube-like crystals. Moreover, the size of the particles increases to 200–600 nm. Interestingly, these particles cluster together by forming numerous aggregates with a size of about 2 μm and such aggregation process can be observed during the whole duration of the reaction. By increasing the reaction time, the crystal particles exhibit more pronounced edges and corners. In addition, the size of the aggregates increases significantly. The size of the clusters in S10, after a crystallization time of 30 min, measures 5 μm and exhibits a spherical structure. By further increasing the time to 120 min and then to 180 min, S12 shows a larger size than S4 (5 μm versus 20 μm), although they were both formed via the aggregation of completely crystallized CHA crystal particles. This observation suggests that the

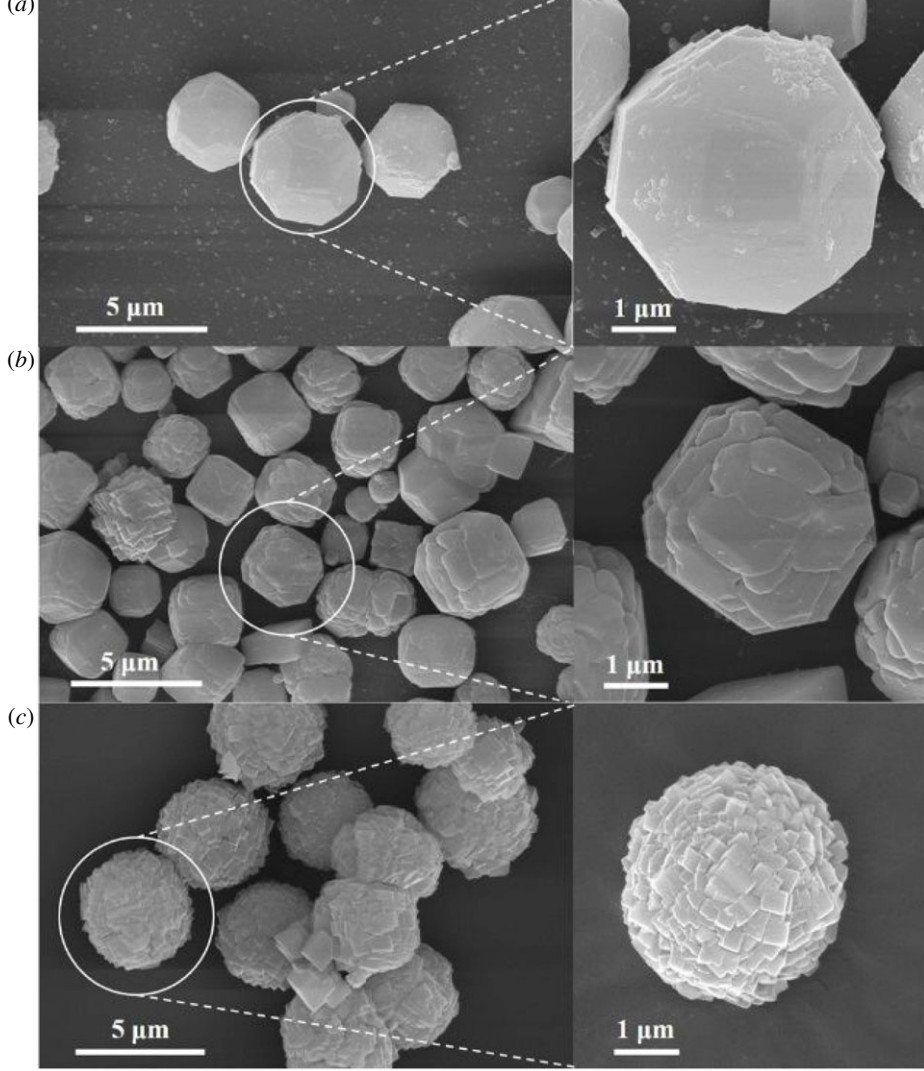

**Figure 9.** SEM images collected at different ageing times and magnified image of the area highlighted by the circle. (*a*) Sample S13; (*b*) Sample S14; (*c*) Sample S4.

increase in the crystallization time not only promotes the crystallinity of the products but also increases the size of the aggregates.

## 3.4. Effects of the ageing time on the sample structure

Sample ageing is a commonly used procedure in the hydrothermal synthesis of molecules and it has been proven to directly affect their crystal character, size and morphology. These properties, to some extent, play a pivotal role in the generation of inter-crystalline mesopores, which generate the cluster structures. To prove this assumption and to explore the influence of the ageing time on the samples, a batch of products were crystallized with different ageing times in the 0–60 min range.

The SEM images of the specimens with different ageing times are reported in figure 9; the samples show a completely different morphology. S13 (figure 9*a*) was prepared without the ageing procedure and it presents a bulk-like morphology with a size of around 4 µm. The rhombic crystals seem to have fostered the aggregation of this structure, which shows a typical CHA type morphology independently of the synthesis process used (i.e. ionothermal or hydrothermal). The sample shape changes into a sphere with diameter of approximately 3–5 µm and with a relatively smooth surface (figure 9*b*) after a 30 min ageing. From the magnified view of the sample, a resemblance with the previous sphere-like morphology can still be observed. By increasing the ageing time, the surface of the crystals gradually becomes uneven. Moreover, numerous crystal particles aggregate to form a bulk

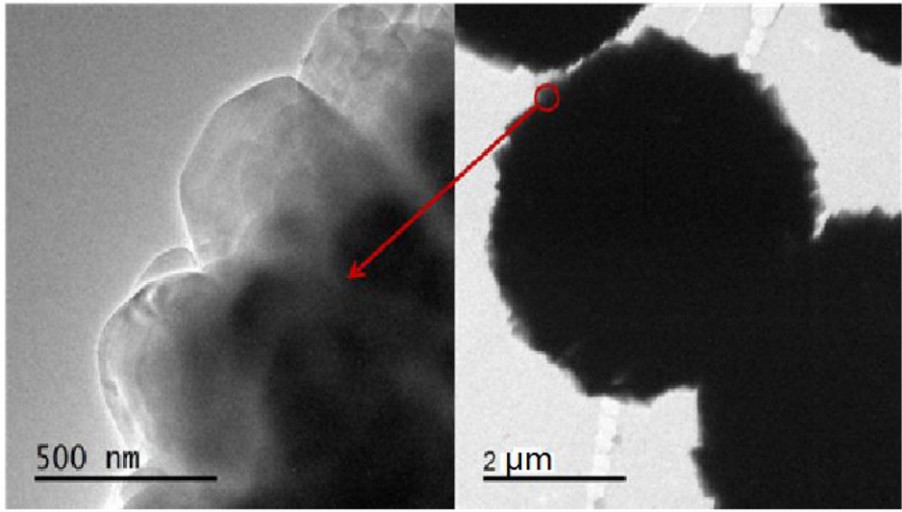

**Figure 10.** TEM images of Sample S4 with high resolution.

**Table 2.** $N_2$ adsorption/desorption isotherms of the S13, S14 and S4 samples.

| sample | BET surface area[a] ($m^2 g^{-1}$) | micropore area[b] ($m^2 g^{-1}$) | external surface area[c] ($m^2 g^{-1}$) | average pore width[d] (nm) |
|---|---|---|---|---|
| S13 | 218.30 | 178.76 | 39.55 | 9.19 |
| S14 | 459.66 | 408.66 | 50.99 | 5.67 |
| S4 | 602.52 | 546.66 | 55.86 | 5.31 |

[a]Calculated by BET method.
[b]Calculated by *t*-plot method.
[c]Calculated by *t*-plot method.
[d]Calculated by Barrett–Joyner–Halenda (BJH) method.

structure. When the ageing time reaches 60 min, the products S4 consist mainly of sphere-like aggregates, which are composed of numerous homogeneous crystal particles with a CHA morphology and clear crystal edges. In addition, the crystal particles with an ageing time of 60 min exhibit a smaller and more uniform size than the products crystallized by using a shorter ageing time. Therefore, the ageing duration efficiently reduces the size of the crystal particles, and sphere-like AlPO-34 molecular sieves are formed upon their aggregation. The TEM image of the Sample S4 is shown in figure 10, and uniform sphere-like nanoparticles are observed in the picture, which is positive to the high surface area and adsorption performance.

The calcined S13, S14 and S4 samples, which exhibit a sphere-like morphology, were selected to perform a deeper investigation of their structural properties and porous parameters via the $N_2$ adsorption/desorption test. The textural parameters of these structures are listed in table 2. The S4 sample has a significantly larger BET surface area ($602.52 \, m^2 g^{-1}$) when compared with S14 and S13 ($459.66$ and $218.30 \, m^2 g^{-1}$, respectively). In addition, its external surface area ($55.86 \, m^2 g^{-1}$) exhibits also the largest extension (S14 $50.99 \, m^2 g^{-1}$ and S13 $39.54 \, m^2 g^{-1}$). This result illustrates that the ageing process affects the textural properties of the AlPO-34 molecular sieves. Relatively long ageing time effectively reduces the size of the crystal particles and the smaller ones can easily assemble into a sphere-like morphology triggered by the LBL self-assembling process.

## 3.5. Adsorption studies on heavy metal ions

The hierarchical AlPO-34 microspheres S4 sample, which exhibits the highest crystallinity, the largest BET surface area, and most abundant channel network among the specimens, was adopted in this study as an adsorbent for heavy metal ions. Its performance in binding to a series of metal cations is shown in figure 11. The ability of the hierarchical AlPO-34 adsorbent to remove Cr(III), Zn(II), Pb(II) and Cu(II)

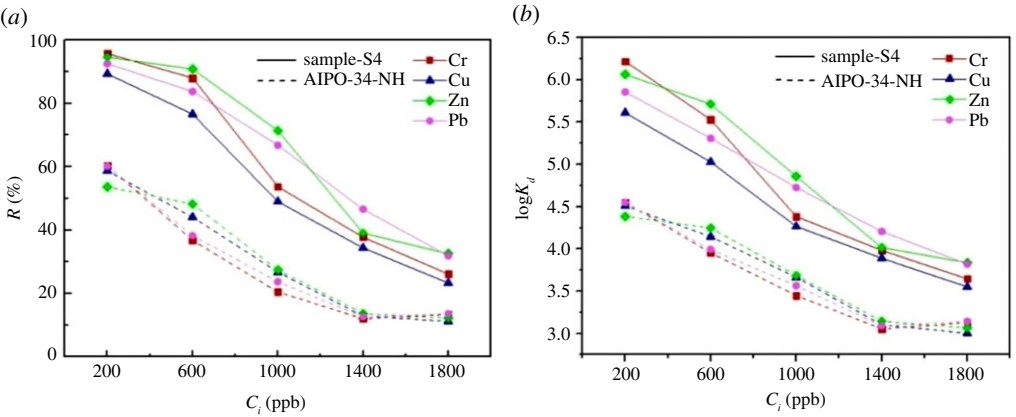

**Figure 11.** Removal rates of different heavy metals (*a*) and distribution coefficients (*b*).

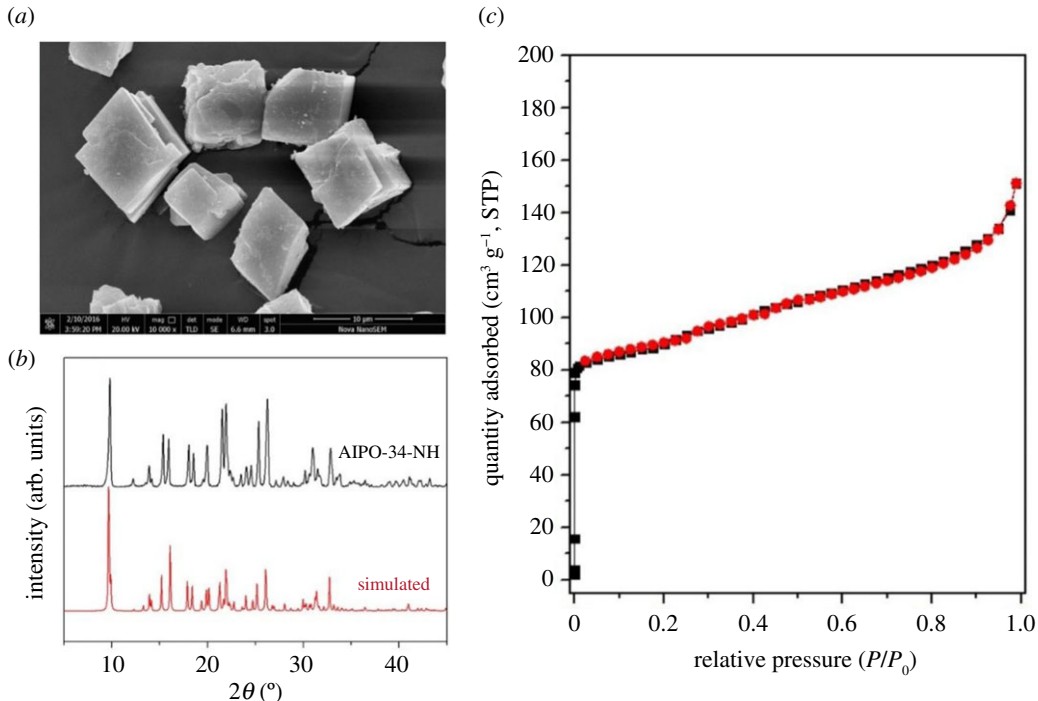

**Figure 12.** (*a*), SEM of AlPO-34-NH; (*b*), XRD patterns of AlPO-34-NH (top) and simulated (bottom); (*c*), N$_2$ adsorption isotherm and desorption isotherm curves of sample AlPO-34-NH.

cations form their respective aqueous solutions is illustrated in figure 11*a*. The majority of the heavy metal cations (greater than 80%) is removed from the solutions. The curves demonstrate that the hierarchical AlPO-34 adsorbent has a relatively higher loading capacity; most of the heavy metal ions in the solution could be removed, leading to the stable percentage, when the cation concentration is low (200 ppb). The removal rates in the AlPO-34-NH adsorbent are approximately only 60% of the hierarchical S4. When the ions concentration got higher (1800 ppb), the adsorption ability of the two adsorbents both decreased.

Equation (2.2) was used to determine such distribution coefficient and the results are illustrated in figure 11*b*. Hierarchical AlPO-34 shows a high $K_d$ value independently of the type of heavy metal cation (Cr $2.21 \times 10^4$ ml g$^{-1}$, Zn $1.73 \times 10^4$ ml g$^{-1}$, Pb $1.24 \times 10^4$ ml g$^{-1}$, Cu $8.34 \times 10^3$ ml g$^{-1}$) when the cations are present in a very low concentration (200 ppb). It can be found that the $K_d$ values of hierarchical adsorbent (Sample-S4) from the type of heavy metal cations are almost twice that of AlPO-34-NH adsorbent. It is supposed that the present synthesized hierarchical adsorbents have better adsorption performance in treatment of heavy metal cations in the water, than that of AlPO-34 molecular sieves without hierarchical structure.

From these significant comparisons, a higher BET surface and a more uniform size of molecular sieve can endow the adsorbent better removal ability (figure 12a,b). Besides, mesoporous with uniform porous width plays an important role in enhancing the adsorption ability of molecular sieves, since the sole present of microporous in AlPO-34-NH adsorbent bring a higher diffusion resistance of the metal cation (figure 12c).

# 4. Conclusion

In conclusion, hierarchical AlPO-34 molecular sieve with a microsphere structure was prepared via the ionothermal synthesis method. The sample has a significantly high BET surface area ($602.52\ m^2\ g^{-1}$). The formation of the hierarchically spherical structure is attributed to the spontaneous agglomeration process of the ionic liquid and the self-assembly process of small-sized crystal particles during the ageing process. During the synthesis, [BMIm]Br served as both the solution and the sole producer of the structure-directing agent. Hierarchical AlPO-34 microsphere adsorbents showed higher efficiency in the treatment of the heavy metal ions in water compared with AlPO-34-NH. Hierarchical AlPO-34 shows a high $K_d$ value independently from the type of heavy metal cation (Cr $2.21 \times 10^4\ ml\ g^{-1}$, Zn $1.73 \times 10^4\ ml\ g^{-1}$, Pb $1.24 \times 10^4\ ml\ g^{-1}$, Cu $8.34 \times 10^3\ ml\ g^{-1}$) and over 80% of the cations could be removed when the cation concentration is low (200 ppb).

Data accessibility. The data are provided in electronic supplementary material [27].

Authors' contributions. H.L. carried out the molecular lab work, participated; L.Z. conceived of the study, designed the study; R.H. carried out the statistical analyses and critically revised the manuscript. All authors gave final approval for publication and agree to be held accountable for the work performed therein.

Competing interests. We declare we have no competing interests.

Funding. We received no funding for this study.

Acknowledgements. The authors wish to thank Natural Science Fund of China (no. 21706022), the Science & Technology Program of Jiangxi Provincial Education Bureau (no. GJJ190557) and Jiangxi Provincial Natural Science Foundation of China (no. 20202BABL203021) for financial support.

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
