## [Peer Review File · Royal Society Open Science]

Review History

RSOS-201322.R0 (Original submission)

Review form: Reviewer 1

Is the manuscript scientifically sound in its present form?

Yes

Are the interpretations and conclusions justified by the results?

Yes

Is the language acceptable?

Yes

Do you have any ethical concerns with this paper?

No

Have you any concerns about statistical analyses in this paper?

No

Recommendation?

Major revision is needed (please make suggestions in comments)

Comments to the Author(s)

The contents of the paper is interesting and well written.

However, I have few observations to modify the paper as given below:

1. Mention what is full form of CHA type in text or in Table-1.
2. In characterization method, TEM analysis is required to determine the size of particles.
3. About different samples S1, S2, S3 ..., the variation of respective parameters may be written clearly in the experimental section to make the reader easy to follow the work. Experimentation method should be written more clearly.
4. Why in Section 2.2, name is "Absorption Experiments"? It should be adsorption instead of absorption.
5. Equation-1 may be written correctly.
6. What is Kd? Explain the equation with kd properly. Adsorption equilibrium constant may be the valid parameter, which is not determined in this paper. Adsorption equilibrium may be incorporated.
7. Why crystallization temperature changes the crystallinity of the structure?
8. What is main reason for changing particle size with crystallization time?
9. Statistical analysis in any form or error bar in the adsorption experiments may be added.

Review form: Reviewer 2

Is the manuscript scientifically sound in its present form?

No

Are the interpretations and conclusions justified by the results?

No

Is the language acceptable?

No

Do you have any ethical concerns with this paper?

No

Have you any concerns about statistical analyses in this paper?

No

Recommendation?

Major revision is needed (please make suggestions in comments)

Comments to the Author(s)

In this manuscript authors have reported hierarchically porous AlPO-34 materials in the presence of [BMIm]Br ionic liquid during synthesis and used them for the adsorptive removal of Cr(III), Zn(II), Pb(II) and Cu(II) from contaminated water. Although results have merit, several areas need improvements before this manuscript can be considered for publication. See the comments below:

- (1) There is some language error in the title, it should be replaced by 'Synthesis of self-assembled hierarchical AlPO-34 microspheres by using an ionic liquid and its application in heavy metal removal'

(2) Introduction should be more comprehensive. Before coming to the specific example of zeolites authors should provide a brief outlook of different porous materials that are used for the removal heavy metal ions. See and include these works: ACS Sustainable Chem. Eng., 2019, 7, 7353-7361; Chem. Commun., 2020, 56, 3963-3966.

(3) Authors should provided the TEM images of their AlPO-34 samples. Without TEM analysis hierarchical nanostructure can't be explained properly.

(4) N₂ sorption isotherms as shown in Figure 10 are not properly explained. All isotherms showed very sharp capillary uptake at low P/P₀ corresponding to the micropores (type I isotherm) in addition to high pressure desorption hysteresis. BJH pore size distribution is not enough to explain these type of isotherms. NLDFT method is more suitable. Authors should get bimodal pore size distributions corresponding to micropores and mesopores. Micropores are not seen in either of these PSDs.

(5) Figure captions are too brief. For example, Figure 12, what is simulated pattern? The isotherm c is not only adsorption isotherm, desorption isotherm is also present there.

Decision letter (RSOS-201322.R0)

This year has been very difficult for everyone, and we want to take the opportunity to thank you for your continued support in 2020.

The Royal Society Open Science editorial office will be closed from the evening of Friday 18 December 2020 until Monday 4 January 2021. We will not be responding during this time. If you have received a deadline within this time period, please contact us as soon as possible to allow us to extend the deadline. If you receive any automated messages during this time asking you to meet a deadline, we offer apologies and invite you to respond after the festive period or during normal working hours.

With our best for a peaceful festive period and New Year, and we look forward to working with you in 2021.

Dear Dr Han:

Title: Self-assembly synthesis of hierarchical AlPO-34 microspheres via an ionic liquid and application in heavy metal removal
Manuscript ID: RSOS-201322

The editor assigned to your manuscript has now received comments from reviewers. We would like you to revise your paper in accordance with the referee and Subject Editor suggestions which can be found below (not including confidential reports to the Editor). Please note this decision does not guarantee eventual acceptance.

Please submit your revised paper before 15-Jan-2021. Please note that the revision deadline will expire at 00.00am on this date. If we do not hear from you within this time then it will be assumed that the paper has been withdrawn. In exceptional circumstances, extensions may be

possible if agreed with the Editorial Office in advance. We do not allow multiple rounds of revision so we urge you to make every effort to fully address all of the comments at this stage. If deemed necessary by the Editors, your manuscript will be sent back to one or more of the original reviewers for assessment. If the original reviewers are not available we may invite new reviewers.

On behalf of the Subject Editor Professor Anthony Stace and the Associate Editor Dr Darren Walsh.

RSC Associate Editor:
Comments to the Author:
(There are no comments.)

RSC Subject Editor:
Comments to the Author:
(There are no comments.)

Reviewers' Comments to Author:
Reviewer: 1

Comments to the Author(s)
The contents of the paper is interesting and well written.
However, I have few observations to modify the paper as given below:
1. Mention what is full form of CHA type in text or in Table-1.

2. In characterization method, TEM analysis is required to determine the size of particles.
3. About different samples S1, S2, S3 ..., the variation of respective parameters may be written clearly in the experimental section to make the reader easy to follow the work. Experimentation method should be written more clearly.
4. Why in Section 2.2, name is "Absorption Experiments"? It should be adsorption instead of absorption.
5. Equation-1 may be written correctly.
6. What is K_d ? Explain the equation with k_d properly. Adsorption equilibrium constant may be the valid parameter, which is not determined in this paper. Adsorption equilibrium may be incorporated.
7. Why crystallization temperature changes the crystallinity of the structure?
8. What is main reason for changing particle size with crystallization time?
9. Statistical analysis in any form or error bar in the adsorption experiments may be added.

Reviewer: 2

Comments to the Author(s)

In this manuscript authors have reported hierarchically porous AlPO-34 materials in the presence of [BMIm]Br ionic liquid during synthesis and used them for the adsorptive removal of Cr(III), Zn(II), Pb(II) and Cu(II) from contaminated water. Although results have merit, several areas need improvements before this manuscript can be considered for publication. See the comments below:

(1) There is some language error in the title, it should be replaced by 'Synthesis of self-assembled hierarchical AlPO-34 microspheres by using an ionic liquid and its application in heavy metal removal'

(2) Introduction should be more comprehensive. Before coming to the specific example of zeolites authors should provide a brief outlook of different porous materials that are used for the removal heavy metal ions. See and include these works: ACS Sustainable Chem. Eng., 2019, 7, 7353-7361; Chem. Commun., 2020, 56, 3963-3966.

(3) Authors should provided the TEM images of their AlPO-34 samples. Without TEM analysis hierarchical nanostructure can't be explained properly.

(4) N₂ sorption isotherms as shown in Figure 10 are not properly explained. All isotherms showed very sharp capillary uptake at low P/P₀ corresponding to the micropores (type I isotherm) in addition to high pressure desorption hysteresis. BJH pore size distribution is not enough to explain these type of isotherms. NLDFT method is more suitable. Authors should get bimodal pore size distributions corresponding to micropores and mesopores. Micropores are not seen in either of these PSDs.

(5) Figure captions are too brief. For example, Figure 12, what is simulated pattern? The isotherm c is not only adsorption isotherm, desorption isotherm is also present there.

Author's Response to Decision Letter for (RSOS-201322.R0)

See Appendix A.

RSOS-201322.R1 (Revision)

Review form: Reviewer 2

Is the manuscript scientifically sound in its present form?

Yes

Are the interpretations and conclusions justified by the results?

Yes

Is the language acceptable?

Yes

Do you have any ethical concerns with this paper?

No

Have you any concerns about statistical analyses in this paper?

No

Recommendation?

Accept as is

Comments to the Author(s)

Authors have addressed the referee comments properly in this revised version of the manuscript and it has been improved. Now this revised manuscript may be accepted in its present form.

Decision letter (RSOS-201322.R1)

Dear Dr Han:

Title: Synthesis of self-assembled hierarchical AlPO-34 microspheres by using an ionic liquid and its application in heavy metal removal

Manuscript ID: RSOS-201322.R1

It is a pleasure to accept your manuscript in its current form for publication in Royal Society Open Science. The chemistry content of Royal Society Open Science is published in collaboration with the Royal Society of Chemistry.

Please see the Royal Society Publishing guidance on how you may share your accepted author manuscript at <https://royalsociety.org/journals/ethics-policies/media-embargo/>. After publication, some additional ways to effectively promote your article can also be found here

<https://royalsociety.org/blog/2020/07/promoting-your-latest-paper-and-tracking-your-results/>.

On behalf of the Subject Editor Professor Anthony Stace and the Associate Editor Dr Darren Walsh.

RSC Associate Editor:
Comments to the Author:
(There are no comments.)

RSC Associate Editor:
Comments to the Author:
(There are no comments.)

Reviewer(s)' Comments to Author:
Reviewer: 2

Comments to the Author(s)
Authors have addressed the referee comments properly in this revised version of the manuscript and it has been improved. Now this revised manuscript may be accepted in its present form.

Appendix A

Reviewers' Comments to Author:

Reviewer: 1

Comments to the Author(s)

The contents of the paper is interesting and well written.

However, I have few observations to modify the paper as given below:

1. Mention what is full form of CHA type in text or in Table-1.

Response: It has been changed in the text. The full name is "chabazite".

2. In characterization method, TEM analysis is required to determine the size of particles.

Response: In order to determine the size of particles. We added the TEM picture of the zeolite.

3. About different samples S1, S2 , S3 ..., the variation of respective parameters may be written clearly in the experimental section to make the reader easy to follow the work. Experimentation method should be written more clearly.

Response: It has been revised according to the suggestion.

4. Why in Section 2.2 , name is "Absorption Experiments"? It should be adsorption instead of absorption.

Response: We are sorry for the mistake. It has been revised.

5. Equation-1 may be written correctly.

Response: It has been changed. It should be " $R=(C_i-C_f)/C_i \times 100/\%$ ".

6. What is K_d ? Explain the equation with k_d properly. Adsorption equilibrium constant may be the valid parameter, which is not determined in this paper. Adsorption equilibrium may be incorporated.

Response: The equaton is changed and adsorption equilibrium is incorporated in the revised paper. The distribution coefficient (K_d , mL/g) is a parameter, which describes the adsorption behavior of the heavy metal cations in aqueous solutions.

7. Why crystallization temperature changes the crystallinity of the structure?

Response: The particle sizes and crystallinity of the products are increased with increase of treatment temperature. According to the XRD of the products, sample S4 prepared with high temperature showed high crystallinity.

8. What is main reason for changing particle size with crystallization time ?

Response: It has been observed that the particle size gradually increased as the synthesis time prolonged, and indicating that the particle size was greatly affected by the synthesis time as shown in Fig.8. At the first step, the crystal formed and grew gradually. After 120 min, no obvious change appears either in the diffraction peaks or in the half-peak width and this demonstrates that the sample crystallizes completely in 120 min. The ionothermal synthesis significantly shortens the crystallization time and enhances the efficiency in the preparation of the molecular sieves.

9. Statistical analysis in any form or error bar in the adsorption experiments may be added.

Response: Adsorption experiment was carried out on each solution with different initial concentrations of metal ions. Each measurement was carried out 3 times in the same conditions. The relation standard deviation of the data was lower than 15%.

Reviewer: 2

Comments to the Author(s)

In this manuscript authors have reported hierarchically porous AIPO-34 materials in the presence of [BMIm]Br ionic liquid during synthesis and used them for the adsorptive removal of Cr(III), Zn(II), Pb(II) and Cu(II) from contaminated water. Although results have merit, several areas need improvements before this manuscript can be considered for publication. See the comments below:

(1) There is some language error in the title, it should be replaced by 'Synthesis of self-assembled hierarchical AIPO-34 microspheres by using an ionic liquid and its application in heavy metal removal'

Response: It has been revised according to the suggestion.

(2) Introduction should be more comprehensive. Before coming to the specific example of zeolites authors should provide a brief outlook of different porous materials that are used for the removal heavy metal ions. See and include these works: ACS Sustainable Chem. Eng., 2019, 7, 7353-7361; Chem. Commun., 2020, 56, 3963-3966.

Response: It has been revised according to the suggestion.

(3) Authors should provided the TEM images of their AIPO-34 samples. Without TEM analysis hierarchical nanostructure can't be explained properly.

Response: We added the TEM picture of the zeolite in the revised paper.

(4) N₂ sorption isotherms as shown in Figure 10 are not properly explained. All isotherms showed very sharp capillary uptake at low P/P₀ corresponding to the micropores (type I isotherm) in addition to high pressure desorption hysteresis. BJH pore size distribution is not enough to explain these type of isotherms. NLDFT method is more suitable. Authors should get bimodal pore size distributions corresponding to micropores and mesopores. Micropores are not seen in either of these PSDs.

Response: N₂ sorption isotherms have been deleted in the revised manuscript. The pore parameters are shown in Table 2.

(5) Figure captions are too brief. For example, Figure 12, what is simulated pattern? The isotherm c is not only adsorption isotherm, desorption isotherm is also present there.

Response: It has been changed like this: Figure 12 (a), SEM of AIPO-34-NH; (b), XRD patterns of AIPO-34-NH (top) and simulated (bottom); (c), N₂ adsorption isotherm and desorption isotherm curves of sample AIPO-34-NH.